# Fecal Transplant and *Bifidobacterium* Treatments Modulate Gut *Clostridium* Bacteria and Rescue Social Impairment and Hippocampal BDNF Expression in a Rodent Model of Autism

**DOI:** 10.3390/brainsci11081038

**Published:** 2021-08-05

**Authors:** Sameera Abuaish, Norah M. Al-Otaibi, Turki S. Abujamel, Saleha Ahmad Alzahrani, Sohailah Masoud Alotaibi, Yasser A. AlShawakir, Kawther Aabed, Afaf El-Ansary

**Affiliations:** 1Department of Basic Sciences, College of Medicine, Princess Nourah Bint Abdulrahman University, P.O. Box 84428, Riyadh 11671, Saudi Arabia; syabuaish@pnu.edu.sa; 2Department of Biology, College of Science, Princess Nourah Bint Abdulrahman University, P.O. Box 84428, Riyadh 11671, Saudi Arabia; noramajd22@gmail.com (N.M.A.-O.); salehah1416@gmail.com (S.A.A.); sohailah1997@gmail.com (S.M.A.); 3Vaccines and Immunotherapy Unit, King Fahd Medical Research Center, King Abdulaziz University, Jeddah 21589, Saudi Arabia; tabujamel@kau.edu.sa; 4Department of Medical Laboratory Technology, Faculty of Applied Medical Sciences, King Abdulaziz University, Jeddah 21589, Saudi Arabia; 5Prince Naif for Health Research Center, King Saud University, P.O. Box 7805, Riyadh 11472, Saudi Arabia; yalshawakir@ksu.edu.sa; 6Central Laboratory, Female Center for Medical Studies and Scientific Section, King Saud University, P.O. Box 22452, Riyadh 11472, Saudi Arabia; elansary@ksu.edu.sa

**Keywords:** autism spectrum disorder, microbiota, propionic acid, *Clostridium perfringens*, *Clostridium* cluster IV, fecal transplant, *Bifidobacterium*, hippocampus, BDNF

## Abstract

Autism is associated with gastrointestinal dysfunction and gut microbiota dysbiosis, including an overall increase in *Clostridium.* Modulation of the gut microbiota is suggested to improve autistic symptoms. In this study, we explored the implementation of two different interventions that target the microbiota in a rodent model of autism and their effects on social behavior: the levels of different fecal *Clostridium* spp., and hippocampal transcript levels. Autism was induced in young Sprague Dawley male rats using oral gavage of propionic acid (PPA) for three days, while controls received saline. PPA-treated animals were divided to receive either saline, fecal transplant from healthy donor rats, or *Bifidobacterium* for 22 days, while controls continued to receive saline. We found that PPA attenuated social interaction in animals, which was rescued by the two interventions. PPA-treated animals had a significantly increased abundance of fecal *C. perfringens* with a concomitant decrease in *Clostridium* cluster IV, and exhibited high hippocampal *Bdnf* expression compared to controls. Fecal microbiota transplantation or *Bifidobacterium* treatment restored the balance of fecal *Clostridium* spp. and normalized the level of *Bdnf* expression. These findings highlight the involvement of the gut–brain axis in the etiology of autism and propose possible interventions in a preclinical model of autism.

## 1. Introduction

Autism spectrum disorder (ASD) is a complex neurodevelopmental disorder that disturbs brain development. The reported incidence of ASD has increased dramatically over the past decade to 1 in 54 births, with males having a higher prevalence than females with a ratio of 4:1 [1]. The etiological mechanisms of ASD remain unclear, although the heterogeneity between patient cohorts supports a complex interaction between environmental and genetic factors [2]. One factor that has received much attention in these past few years is gut microbiota dysbiosis as an underlying link to understanding ASD etiology. Clinical data show an imbalance in the microbiota of autistic individuals compared to neurotypical ones. Studies suggest a significant imbalance between the *Bacteroidetes* and *Firmicutes* phyla in ASD [3,4,5]. For example, children with ASD exhibited less diverse microbiota with lower *Bifidobacterium* and *Firmicutes* and higher *Lactobacillus*, *Clostridium*, *Bacteroidetes*, *Desulfovibrio*, *Sarcina*, and *Caloramator* levels compared to children without ASD [3]. 

The *Clostridium* genus is a predominant cluster of gut microbiota and classified as anaerobic, Gram-positive, rod-shaped, and spore-forming bacteria. More recently, following the sequencing of 16S rRNA, the diversity between *Clostridium* spp. became more noteworthy, as more than 200 variants of *Clostridium* spp. were recognized. Depending on their species and strains, the *Clostridium* may exert many benefits or potential risks to the host’s health [6]. For instance, *Clostridium* clusters XIVa and IV, which have a great potential acting as probiotics, were reduced in autistic children [7]. *Clostridium* clusters XIVa and IV play a key role in the normalization of germ-free (GF) mice gut microbiota [8]. Besides, cellular components of *Clostridium* cluster XIVa and IV and their metabolites (e.g., butyrate, secondary bile acids) can exert anti-inflammatory effects and increase gut immune tolerance [9,10]. On the other hand, autistic children with gastrointestinal symptoms had a significantly higher abundance of *Clostridium perfringens* [11,12]. Indeed, *C. perfringens* are virulent bacteria that simulate gastrointestinal dysfunction through the production of more than 20 toxins, as well as antimicrobial and drying-resistant spores [13]. 

The gut–brain axis consists of a communication network that includes the vagal nerve, microbial metabolites, gut hormones, and the endocrine and immune systems, which control the gut process and link it to the brain [14,15]. Illustrating the role of microbiota in the gut–brain axis has been demonstrated in GF mice. These mice revealed abnormalities in hippocampal gene expression and deficits in behavior that were more significant in male mice [16]. ASD-like symptoms in these mice ameliorated after restoring the gut microbial community [17]. In addition, behavior abnormality symptoms in GF mice were partially alleviated after fecal matter transplant and fully ameliorated by the ingestion of probiotics [18]. 

The production of short-chain fatty acids, such as propionic acid (PPA), by the gut *Clostridia* and *Desulfovibrio,* is suggested to play a role in developing ASD [19]. PPA is a by-product of the digestion of high carbohydrate foods by gut bacteria fermentation. In addition, PPA is used as a preservative in refined food products [19]. Interestingly, reports of increased severity of ASD symptoms were associated with the consumption of either food source, while eliminating these products was associated with improvement of symptoms [20,21]. In support of the suggested role of PPA in ASD etiology, PPA treatments, including subcutaneous, intragastric gavage, intraperitoneal, or intracerebroventricular, have induced autistic-like features in rodent models of ASD, including behavioral and physiological traits [22]. 

Abnormalities in the hippocampal structure and neurobiology have been observed in both autistic human and ASD animal models. For example, autistic individuals were reported to have reduced neuronal dendritic branching compared to non-autistic individuals [23]. Moreover, PPA treatment reduced the thickness of the granular layer of the dentate gyrus of the hippocampus, reduced the total number of synaptic vesicles, and induced dendritic spine loss in vitro [22,24,25]. Brain-derived neurotrophic factor (BDNF) is well known for its role in brain development and plasticity. Serum BDNF levels have been reported to be higher in autistic individuals [26]. BDNF levels in the brain of animal models of autism have shown unclear results ranging from no change, increase, or decrease [27,28,29]. Interestingly, BDNF transcript levels have been shown to be regulated by gut microbiota and short-chain fatty acids [30,31]. BDNF levels are also known to be regulated by methyl-CpG binding protein 2 (MeCP2). The MeCP2 gene is mutated in several neurodevelopmental disorders, including autism, and its protein is known to repress BDNF transcriptionally [32]. 

In this study, we evaluated social behavior impairment and the modulation of fecal *Clostridium* spp. levels, specifically *Clostridium* cluster XIVa and IV and *C. perfringens* in a rodent model of autism induced by oral PPA treatment. We also examined hippocampal *Bdnf* and *Mecp2* transcript levels along with solute carrier family 17, member 7 (*Slc17a7*) and glutamate decarboxylase 1 (*Gad1*) transcript levels as markers of glutamatergic and GABAergic neurons. Finally, we implemented an intervention treatment using either a fecal transplant or probiotic treatment of *Bifidobacterium*, which showed rescuing effects on social behavior, fecal *Clostridium* levels, and hippocampal *Bdnf* transcript levels in this PPA-induced animal model of autism. This study might highlight the benefits of modulating gut microbiota through different interventional modalities and its link to molecular neurobiology in the context of autism. 

## 2. Materials and Methods

### 2.1. Animals 

Twenty-eight Sprague Dawley male rats weighing 80–120 g and aged approximately 28 days old were acquired from in-house breeding from different litters from the Center for Laboratory Animals and Experimental Surgery at King Saud University (Riyadh, Saudi Arabia). Animals were housed in groups of 3–4 animals per cage at 21 ± 1 °C and exposed to 12:12 h light–dark cycle and had access to food and water ad libitum. The Ethics Committee approved all procedures for Animal Research of King Saud University, Riyadh (IRB No: KSU-SE-19-61) and the Research Ethics Committee of Princess Nourah bint Abdulrahman University, Riyadh (IRB No.19-0103).

### 2.2. Experimental Design 

On the first day of testing, rats were designated to receive either 1 mL oral saline (control *n* = 7) or an oral dose of 250 mg/kg PPA (*n* = 21) dissolved in distilled water for three days [33]. Later, the PPA-treated animals were divided to receive either saline (PPA *n* = 7), fecal transplant (FT *n* = 7), or *Bifidobacteria* (BF *n* = 7) for 22 days. For the fecal transplant group, 1 g of pooled fecal samples from healthy donor rats was suspended in 10 mL of sterile phosphate-buffered saline (PBS, pH 7.4; Gibco^TM^, Thermo Fisher Scientific Inc, Waltham, MA, USA) by vortexing. The homogenized solution was later filtered twice through a sterilized metal sieve. Fecal transplantation was performed by oral gavage of the fecal filtrate at a dose of 1 g/kg [34]. For the *Bifidobacteria* (BF)-treated group, powder from 1 capsule of *Bifidobacterium longum* BB536 (Bifido GI balance, Life Extension), which contains 2 × 10^9^ colony forming units (CFU) per 25 mg, was dissolved in 1 mL sterile PBS. Animals were given 0.5 mL daily (1 × 10^9^ CFU) by oral gavage [35]. The control animals continued to receive oral saline for 22 days. Freshly evacuated fecal pellets were collected weekly into sterile microtubes and stored at −80 °C until later assayed. After the last day of treatment, animals were handled daily for 2 min for a week. Behavioral testing was conducted, and a day later, animals were euthanized, and their brains were harvested and dissected (Figure 1).

### 2.3. Social Behavior

Social behavior was assessed using the three-chamber social test, a commonly used test to evaluate social impairments in animal models of autism [36,37]. An 80 cm × 40 cm × 40 cm clear rectangular plexiglass box that was divided into three chambers with walls with 15 cm × 15 cm doors with removable sliders in order to allow the animals to pass through was used for the test. The three-chamber box was cleaned with 70% ethanol, dried with paper towels, and then let to air dry between trails. The animals were transferred to the testing room one hour before the test for acclimatization. Animals were picked from the cage and placed individually in the center chamber and were allowed to explore for 5 min while the two doorways of the box were closed. Following this habituation period, both doors were opened immediately, and a novel same-sex conspecific rat with similar body weight was placed in one of the two perforated holding containers that were located on either side of the box. The subject rat was allowed to explore all three chambers freely for 10 min. The conspecific rat position was alternated between animals to avoid side preference. The test was recorded using an HD camcorder (Legria, Canon, Tokyo, Japan). Videos were later analyzed to code behaviors using BORIS 7.9.16 software [38] with examiners who were blind to animal treatments. Time of social interaction, which was defined by the orientation towards and investigation of the cup holding the conspecific animal by the focal animal through sniffing or rearing against it, was obtained. Moreover, time spent in each chamber was analyzed, and the percent of time spent in the social chamber (containing the conspecific rat) relative to the non-social chamber (containing the empty holding container) and the center chamber was calculated. Time spent immobile (bouts of no movements) was also recorded. 

### 2.4. DNA Extraction and Microbial Quantitative Real-Time PCR 

Metagenomic DNA was extracted from day 11 and day 22 fecal samples (*n* = 5/group) using the QIAamp DNA soil Kit (Qiagen, Hilden, Germany) according to the manufacturer’s instructions. The final DNA concentration and purity were quantified by a spectrophotometer (Nanodrop, Thermo Fisher Scientific Inc., Waltham, MA, USA) and samples were stored at −20 °C until further use. Quantification of microbiota was carried out by qPCR using 7500 Real-Time PCR System (Applied Biosystems, Thermo Fisher Scientific Inc., Waltham, MA, USA). Each PCR reaction was set up in a total volume of 25 μL per reaction using 50 ng of fecal DNA, which was added to a reaction mixture containing 1 μM of each primer and 2× SYBR^®^ master mix (Applied Biosystems). qPCR cycling conditions were performed according to the manufacturer’s recommendation (Applied Biosystems). PCR reactions were performed in duplicate. Target gene ΔCt values were calculated against the Ct value of the universal bacterial 16S rRNA. The fold change of the three experimental groups from the control saline group was calculated by computing 2^−ΔΔCt^. The primers used in the current study are listed in Table 1. 

### 2.5. Brain Collection and RNA Extraction 

On the last day of the study, animals were euthanized by inhalation of an overdose of sevoflurane, and the animals were decapitated using a rodent guillotine. The brain was rapidly removed from the skull and placed on a cold metal dissecting plate placed onto ice. The brain was freshly dissected to collect hippocampal tissues from both hemispheres. Hippocampi were placed into 1.5 mL Eppendorf tubes, flash frozen on dry ice, and stored at −80 °C. 

Brain tissue was homogenized in lysis buffer by passing it through a 20-gauge needle attached to a sterilized plastic syringe at least ten times or until a homogeneous lysate was achieved. RNA was extracted using the Allprep DNA/RNA Micro Kit (Qiagen, Hilden, Germany) according to manufacturer’s instructions. Quantification and purity of the RNA were assessed with a spectrophotometer (Genova nano, Jenway, Cole-Parmer Ltd., Staffordshire, UK). RNA (0.5 μg) was converted into cDNA using the High-Capacity cDNA Conversion Kit (Applied Bioystems, Thermo Fisher Scientific Inc., Waltham, MA, USA) according to manufacturer’s instructions. 

### 2.6. Hippocampal Gene Expression Analysis by Quantitative Real-Time Reverse Transcriptase-Polymerase Chain Reaction (qRT-PCR)

The expression patterns of 4 genes (*Bdnf*, *Mecp2*, *Slc17a7*, and *Gad1*) and three housekeeping genes (*ActinB*, *18S rRNA*, and *Gapdh*) were quantified (*n* = 5/group) and analyzed using QuantStudio 5 Real-Time PCR (Applied Biosystems) with HOT FIREPol SolisGreen qPCR Mix (Solis Biodyne, Tartu, Estonia). The reaction was set up following the manufacturing instructions. Samples were run in triplicates, and the average cycle threshold (Ct) values were calculated. Target gene ΔCt values were calculated using the geometric mean of the three housekeeping genes. The fold change of the three experimental groups from the control saline group was calculated by computing 2^−ΔΔCt^. Primers (Table 1) were designed according to GenBank sequence information at the National Center for Biotechnology Information (NCBI; www.ncbi.nlm.nih.gov).

### 2.7. Statistical Analysis

Prism version 9.1.1 was used for data analysis and visualization; the results were expressed as mean ± SE. The data were checked for normality using the Shapiro–Wilk test. Two-way ANOVA followed by Fisher’s least significant difference post hoc test was used to analyze data for interaction duration in the 3-chamber social test. One-way ANOVA followed by Fisher’s least significant difference post hoc test was used to analyze data for percent of time spent in the social chamber, fold change in *Clostridium perfringens* on day 11, fold change in *Clostridium* cluster IV, fold change in *Clostridium* cluster XIVa on day 22, and fold change in hippocampal gene expression. Kruskal–Wallis non-parametric test followed by uncorrected Dunn’s test for multiple comparisons was used to analyze *Clostridium* cluster IV and *Clostridium* cluster XIVa data on day 11, and *Clostridium perfringens* on day 22. 

## 3. Results

### 3.1. Social Behavior

The different treatments differentially affected the time spent interacting with the two holding containers (F (3, 47) = 3.240, *p* = 0.03; Figure 2A). Overall, animals spent more time interacting with the conspecific animal (F (1, 47) = 10.67, *p* = 0.002). Specifically, saline control animals spent significantly more time interacting with the conspecific animal than time spent with the empty holding container (Fisher’s LSD: *p* = 0.02). On the other hand, PPA animals did not show a significant difference in time spent interacting with the conspecific animal and the empty holding container (Fisher’s LSD: *p* = 0.61). PPA animals also spent significantly less time interacting with the conspecific animal than saline control animals (Fisher’s LSD: *p* = 0.01). FT-treated animals did not show a significant difference between time spent interacting with the conspecific animal and the empty holding container (Fisher’s LSD: *p* = 0.14). However, FT-treated animals showed a trend of more time spent interacting with the conspecific animal compared to PPA-treated animals (Fisher’s LSD: *p* = 0.06). BF-treated animals spent significantly more time interacting with the conspecific animal compared to time spent with the empty holding container (Fisher’s LSD: *p* = 0.03). Moreover, BF-treated animals spent significantly more time interacting with the conspecific animal than PPA-treated animals (Fisher’s LSD: *p* = 0.005). 

Similarly, the different treatments significantly affected the percentage of time spent in the social chamber relative to the other chambers (F (3, 23) = 4.46, *p* = 0.01; Figure 2B). Specifically, PPA treatment significantly reduced the percentage of time spent in the social chamber when compared to control saline animals (Fisher’s LSD: *p* = 0.005). FT treatment increased the percentage of time spent in the social chamber compared to PPA treatment, but it did not reach statistical significance (Fisher’s LSD: *p* = 0.15). On the other hand, BF treatment significantly increased the percentage of time spent in the social chamber compared to PPA treatment (Fisher’s LSD: *p* = 0.006).

In addition, we found a main effect of treatment on time spent immobile (bouts of no movements) by animals (F (3, 23) = 4.182, *p* = 0.02; Figure 2C). Specifically, PPA treatment increased the immobility of animals compared to saline (Fisher’s LSD: *p* = 0.008), which was rescued by the two interventions (FT: Fisher’s LSD: *p* = 0.04; BF: Fisher’s LSD: *p* = 0.004).

### 3.2. Fecal Clostridium Bacteria Levels

On day 11, the different treatments affected the levels of Clostridium perfringens (F (3, 11) = 5.94, *p* = 0.01; Figure 3A), with the PPA animals exhibiting higher levels compared to all groups (Fisher’s LSD: control saline *p* = 0.004; FT *p* = 0.009; BF *p* = 0.003). Clostridium cluster IV levels (Figure 3B) were significantly different between the different treatment groups (X^2^ (3) = 11, *p* = 0.01), with PPA animals exhibiting significantly lower levels compared to the other groups (uncorrected Dunn’s tests: control saline *p* = 0.007; FT *p* = 0.02; BF *p* = 0.003). Clostridium cluster XIVa levels (Figure 3B) were comparable among the different treatment groups (X^2^ (3) = 3.17, *p* = 0.36). 

On day 22, only Clostridium cluster IV levels (Figure 3E) were significantly different between the different treatment groups (F (3, 16) = 3.269, *p* = 0.05). BF-treated animals had significantly lower Clostridium cluster IV levels than control saline (Fisher’s LSD: *p* = 0.03) and PPA-treated animals (Fisher’s LSD: *p* = 0.01). There were no differences in the levels of Clostridium perfringens (Figure 3D) and Clostridium cluster XIVa (Figure 3E).

The radar plot shows that, after 11 days of the different treatments, PPA-treated animals show a divergent pattern of levels of the three bacteria compared to the control and the FT and BF intervention treatments, which show comparable levels to the control animals (Figure 4A). On the last day of the treatments (day 22), the patterns of the levels of the three bacteria were similar (Figure 4B). 

### 3.3. Hippocampal Gene Expression

Transcript levels of hippocampal *Bdnf* (Figure 5A) were significantly different between the different treatment groups (F (3, 16) = 4.14, *p* = 0.02). PPA-treated animals exhibited significantly higher levels than the controls (Fisher’s LSD *p* = 0.008) and FT-treated animals (Fisher’s LSD: *p* = 0.007). PPA animals also showed a trend of an increase in *Bdnf* transcript levels compared to BF-treated animals (Fisher’s LSD: *p* = 0.07). In addition, *Mecp2* transcript levels (Figure 5B) were different between the different treatment groups (F (3, 16) = 4.14, *p* = 0.02), with FT-treated animals exhibiting higher levels compared to all other groups (Fisher’s LSD: saline *p* = 0.01; PPA *p* = 0.03; BF *p* = 0.01). Transcript levels of hippocampal *Slc17a7* (Figure 5C) showed a trend of differences among the different treatment groups (F (3, 16) = 2.82, *p* = 0.07). FT-treated animals exhibited higher transcript levels than all other groups (Fisher’s LSD: saline *p* = 0.03; PPA *p* = 0.06; BF *p* = 0.02). *Gad1* transcript levels (Figure 5D) did not significantly differ among the different treatment groups (F (3, 16) = 1.68, *p* = 0.21). 

## 4. Discussion

ASD is a neurodevelopmental disorder that shows an association with differential gut microbiota diversity and metabolites [3,4,5]. Consequently, there are several attempts of interventional treatments to modulate the gut microbiota in order to manage ASD symptoms [42]. In this study, we implemented an interventional paradigm using either fecal microbiota transplantation or probiotic treatment using *Bifidobacterium* in an animal model of autism induced by oral PPA treatment. We found that the implemented interventions successfully improved the social behavior impairment observed in PPA-treated animals. PPA treatment induced a transient increase and decrease in fecal levels of the harmful *Clostridium perfringens* and the probiotic acting *Clostridium* cluster IV, respectively. Finally, the PPA-treated animals exhibited significantly higher hippocampal *Bdnf* transcript levels that were normalized in the two intervention groups. 

Given the strong evidence highlighting the link between neurological diseases and gastrointestinal health, including ASD, a number of interventional methods targeting the gut microbiota have been explored. These interventions include administering fecal transplants, probiotics, or prebiotics [42]. A fecal transplant, or fecal microbiota transplant, defined as administering a fecal solution containing microbiota from a healthy donor to a patient [43], has been reported to improve gastrointestinal and behavioral symptoms and increase gut bacterial abundance and diversity in autistic children [44]. *Bifidobacterium* forms a significant fraction of the human microbiome and is significantly reduced in both autistic individuals and autistic children, while the supplementation of *B. longum* along with other probiotics was found to improve the severity of ASD and reduce gastrointestinal symptoms [45,46,47]. *B. longum* BB536 has been added to many dairy-based products, including infant formula, and is suggested to have multiple health benefits [48]. This study implemented both interventions for three weeks following PPA treatment in order to explore their possible benefit in restoring social behavior, fecal *Clostridium* spp. levels, and hippocampal transcript levels.

First, we evaluated the effects of PPA treatment on social behavior using the three-chamber social test, which utilizes the innate preference of rodents to interact with social stimuli over an inanimate object, and has been applied extensively to assess social impairment in rodent models of autism [36,37]. We found that PPA treatment induced a social behavior deficit, indicated by the loss of the preference to interact with a social stimulus (conspecific animal) (Figure 2A,B). These results indicate long-lasting (>4 weeks) effects of the initial 3-day treatment of PPA. This is in parallel with other studies using PPA treatment, which also report social impairment measured more than four weeks after the initial PPA treatment [22,49,50,51]. PPA animals also exhibited an increased immobility time compared to the control animals (Figure 2C), which might indicate an increased anxiety-like behavior, which is a behavioral feature of ASD models [27,49]. However, validated tests of anxiety need to be performed in future studies to better assess anxiety-like behavior in this model of autism.

Overall, the interventional treatments showed a rescue of the impaired social behavior that was observed in PPA-treated animals (Figure 2). Fecal transplant treatment post-autism-induction improved social behavior; however, the change in behavior did not reach significance. Others have also reported a non-significant trend of improvement of social behavior after fecal transplant treatment from healthy human donors in a maternal immune activation rodent model of autism [52]. It has been suggested that prolonged fecal transplant treatment could be required in order to achieve profound remission in some patients with inflammatory bowel disease [53]. Clinical studies reporting improved ASD symptoms after fecal transplant treatment have typically used a course of treatment that is 7–8 weeks or more [54,55,56]. On the other hand, *B. longum* treatment significantly improved the social behavior in PPA-treated animals. Multiple studies in humans and animal models have reported beneficial effects of *Bifidobacterium* administration on different behavioral outcomes, such as anxiety and depression [57,58,59,60]. There are limited studies examining the impact of *Bifidobacterium* treatment on social behavior. One study has found an increased interaction in mice with an aggressive mouse in a model of chronic social defeat stress after *Bifidobacterium* treatment [61]. Some evidence suggests that *Bifidobacterium* mediates its effects through vagal signaling [62]. Further, others have shown that *Bifidobacterium* administration might rescue behavioral deficits by increasing the synthesis of 5-hydroxytryptamine in vitro and serotonin in the brain [58]. Our interventional treatments show promising results in ameliorating social behavior impairments induced by PPA treatment. 

*Clostridium* is the predominant cluster of commensal bacteria in human and animal intestines, with several species with a wide range of effects [63]. In the present study, after two weeks of the treatment administration, we found that the abundance of *C. perfringens* was significantly elevated in PPA-treated animals compared to the controls (Figure 3A). These findings align with many reports of detecting *C. perfringens* in autistic patients [11,12,64]. It has been previously confirmed that high detection rates of *C. perfringens* and toxins in autistic children are associated with gastrointestinal troubles [12]. Specifically, *C. perfringens* containing the beta-2 toxin gene are found significantly more frequently in autistic individuals compared to controls [11,12,64]. To our knowledge, no studies have examined the effects of the beta-2 toxin on the brain; however, another toxin produced by *C. perfringens* (epsilon) has been reported to induce neural injury in rats and can bind to glial cells and the myelin sheaths of neurons [65,66,67]. More studies examining the effects of the beta-2 toxin on the brain could shed light on its relevance to ASD etiology. 

We also found that PPA-treated rats showed a significant decrease in the abundance of *Clostridium* cluster IV compared to controls (Figure 3B). At the same time, there was no significant difference in the levels of *Clostridium* cluster XIVa (Figure 3C). In a previous study, cultured bacteria from fecal samples obtained from PPA-treated golden hamsters showed *Clostridia* growth, which was absent in the control animals [68]. In addition, treatment of these animals with a probiotic containing a mixture of bacteria, including *Bifidobacteria,* inhibited *Clostridia* growth. More recently, *Clostridium* cluster IV and XIVa reduction have been reported in autistic children [7]. *Clostridium* cluster IV accounts for 16–25% of the fecal microbiota and consists of essential members, including *C. leptum*, *Faecalibacterium prausnitzii* (*F. prausnitzii*), *C. sporosphaeroides*, and *C. cellulosi* [69,70,71]. The health benefits from *Clostridium* cluster IV are attributed to its anti-inflammatory action that inhibits nuclear factor-kappa B (NF-κB) activation and IL-8 production [72]. In addition, these clusters are butyrate-producing bacteria, a short-chain fatty acid that is reportedly reduced in autistic individuals [73]. Butyrate plays a role in maintaining the integrity of the gut epithelial barrier by upregulating the expression of the tight junction-associated protein [74]. Moreover, sodium butyrate administration in an autistic rodent model improves social behavior [75]. 

These differential *Clostridia* levels observed in PPA-treated animals were only transiently observed after approximately two weeks of treatment and were not observed at the terminal time point of treatment (22 days). It appears that PPA-treated animals recovered this transient imbalance in the *Clostridia* levels by the end of the treatment course (Figure 3 and Figure 4). At the first time point, animals in this study were aged around 42 days old and still considered juveniles [76], a developmental period mainly examined in the literature for autism-related microbiota differences [46,77,78]. At the last time point in this study, animals were aged around 52 days, a period in which animals reached sexual maturity and adolescence [76]. Studies have shown that the gut microbiota of rodents changes after puberty and that sex hormones mediate that change in the microbiota [79,80]. Although we did not find effects of PPA treatment at the later time point, we cannot exclude differential effects on other gut microorganisms. A future study will examine the microbiome in this model of autism.

We found that fecal transplant and *B. longum* treatments following PPA treatment were able to normalize *C. perfringens* and cluster IV levels at day 11 of treatment. On day 22, we found that *B. longum*-treated animals had significantly reduced levels of *C.* cluster IV compared to control animals and PPA-treated animals. Fecal transplantation has been used previously in autistic children and was found to increase the abundance of *Bifidobacterium*, *Prevotella*, and *Desulfovibrio* [54,55]. In addition, fecal transplantation has been successfully used to treat recurrent *C. difficile* infection [81], another pathogenic *Clostridium* spp. Our results show beneficial effects of fecal transplant in reducing the pathogenic *C. perfringens* abundance seen in PPA-treated animals. The effects of *Bifidobacterium* treatment in our study are supported by evidence indicating the suppression of *C. perfringens* in vivo and in vitro by *Bifidobacterium* [82,83]. Moreover, *B. longum* BB536 production of acetate during carbohydrate digestion was found to stimulate butyrate-producing bacteria in vitro [84], which could be a mechanism through which it increased *Clostridium* cluster IV abundance after PPA treatment in our study. Our findings highlight the importance of examining different species of *Clostridium* in the context of autism and how they respond to different interventions. 

We finally examined the effects of PPA treatment on transcript expression of *Bdnf*, *Mecp2*, *Slc17a7*, and *Gad1* in the hippocampus. ASD affects the structure and function of several brain regions, including the hippocampus. Studies have reported a decreased dendritic complexity and thickness the hippocampus in humans and animal models that could be related to behavioral abnormalities associated with ASD [22,23,24,25]. The hippocampus connection with the amygdala plays an important role in mediating social interactions as well [85]. We found that PPA treatment alone has significantly increased the expression of *Bdnf* in the hippocampus (Figure 5A). This increase was restored to control levels in both intervention treatment groups (Figure 5A). Studies of animal models of autism have reported increased BDNF protein and mRNA levels in fetal brains [29,86]. In addition, several clinical studies have reported significantly higher levels of BDNF in the serum of autistic individuals [26]. However, others using the subcutaneous injection of PPA in rats have reported a significant reduction in the protein levels of BDNF in the hippocampus in juvenile animals [27]. Interestingly, a study using a transgenic model of autism (*Shank1* knock-out) reported an increase in *Bdnf* transcript levels in *Shank1* knock-out animals only after an object recognition task and not at baseline, and levels were negatively correlated with object recognition memory [87]. In this study, we have measured the transcript levels in animals after performing the social test during which animals are acquiring information about their environment and learning, which could have influenced the levels of the *Bdnf* transcript, unmasking differences that would not have been recorded at baseline. In addition, some reports indicate that BDNF is upregulated by NF-κB, a primary protein in the neuroinflammatory cascade [88]. Neuroinflammation and enhanced NF-κB levels is a feature that has been reported by many studies of autism [89]. Consequently, the increase in *Bdnf* transcript levels might act as a compensatory mechanism downstream of the neuroinflammation that has been reported previously in the PPA model of autism [19,22]. Our study is limited in just measuring transcript levels and not protein levels, especially given the differential function of the pro-BDNF form (precursor uncleaved BDNF) and the mature form of BDNF in neuronal function [90]. In fact, one study has reported an altered balance in the pro-BDNF/BDNF ratio in autistic human brains [90]. More studies should closely examine the role BDNF plays in the etiology of autism and the relevance of its increased levels in the serum to its function in the brain.

We did not observe any effects of PPA treatment on the other transcripts. However, the fecal transplant treatment has significantly increased the transcript levels of *Mecp2* and *Slc17a7* compared to all other groups (Figure 5B,C). Reduced MeCP2 levels have been found in the pre-frontal cortex of autistic individuals and animal models of autism [91,92], which we did not find in the hippocampus in our model. Fecal transplant treatment might have restored the levels of *Bdnf* transcript by increasing the MeCP2 levels. MeCP2 protein is a known transcriptional suppressor of BDNF [32]. There are no previous reports linking the fecal transplant to the modulation of MeCP2 levels. It is not clear the mechanism by which fecal transplant treatment induced this increase in MeCP2 levels. Examining differential microbiome metabolome in response to the fecal transplant in this model could shed light on possible factors contributing to this effect. The alteration in mice microbiome using GF and specific-pathogen-free mice has recently been reported to influence the expression of several genes involved in the neural excitation–inhibition balance, including *Bdnf* and *Slc17a7* in different brain regions [93]. Previous work has indicated increased glutamate and decreased GABA in the brains of PPA–treated animals [68,94], which is not replicated in this study by measuring *Slc17a7* and *Gad1* transcript levels as markers of the excitation–inhibition balance. The increase observed in *Slc17a7* by fecal transplant treatment without an increase in *Gad1* levels might suggest that there is increased excitatory signaling, which could explain the lack of full recovery of social impairment of this interventional treatment. Further examination of glutamate and GABA levels after fecal transplantation should be considered in order to fully understand its effects on the excitation–inhibition balance in the hippocampus in this model. Overall, our hippocampal transcript results implicate the alteration in *Bdnf* transcript levels in the neuropathology of autism induced by PPA administration, and its modulation by different microbiota-directed interventions, which is possibly associated with social behavioral improvement observed after the interventions. 

## 5. Conclusions

This study explored the implementation of two interventional treatments, fecal transplantation and probiotic treatment using *Bifidobacterium*, that targeted the microbiota of an animal model of autism. We demonstrated that orally administering PPA, a short-chain fatty acid produced by gut microbiota and commonly elevated in autistic children, induced social impairments in animals, which was normalized by the interventions. In addition, here, we examined the differential modulation of harmful and beneficial *Clostridium* spp. in response to PPA induction of autism, rather than examining the overall genus *Clostridium*, which is generally reported in the literature. The interventional treatments restored the imbalance observed in *C. perfringens* and cluster IV after PPA treatment. Finally, we report that the impairment in social behavior and imbalanced *clostridium* spp. induced by PPA treatment was associated with augmented *Bdnf* transcript levels in the hippocampus. The two interventional therapies were successful in reducing the abnormal increase in *Bdnf* levels in the hippocampus. These findings propose targeting the microbiota as a potential intervention in autism and provide possible underlying mechanisms of action in a preclinical model of autism. 

## Figures and Tables

**Figure 1 brainsci-11-01038-f001:**
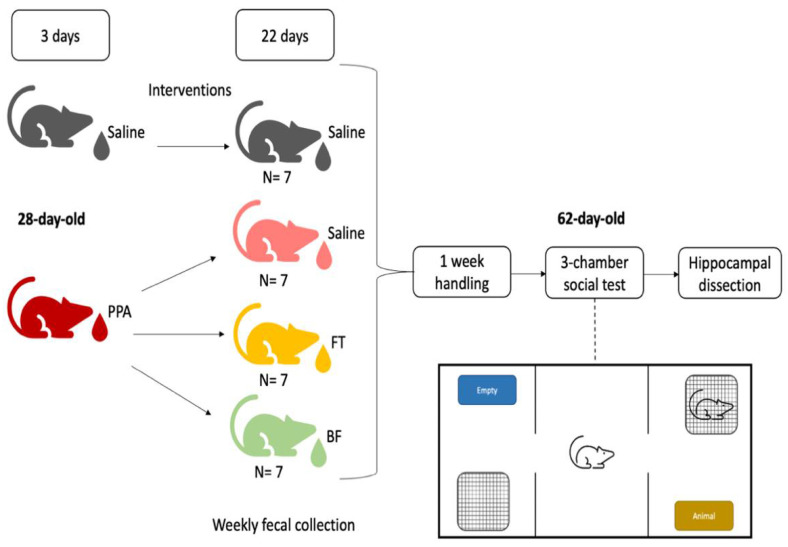
Study experimental design. Animals either received propionic acid (PPA) or saline for three days, after which PPA animals were treated with either saline, fecal transplant (FT), or Bifidobacterium (BF) for 22 days. Later, animals were evaluated for sociability using the 3-chamber social test and sacrificed a day later, their brains collected, and the hippocampi dissected out.

**Figure 2 brainsci-11-01038-f002:**
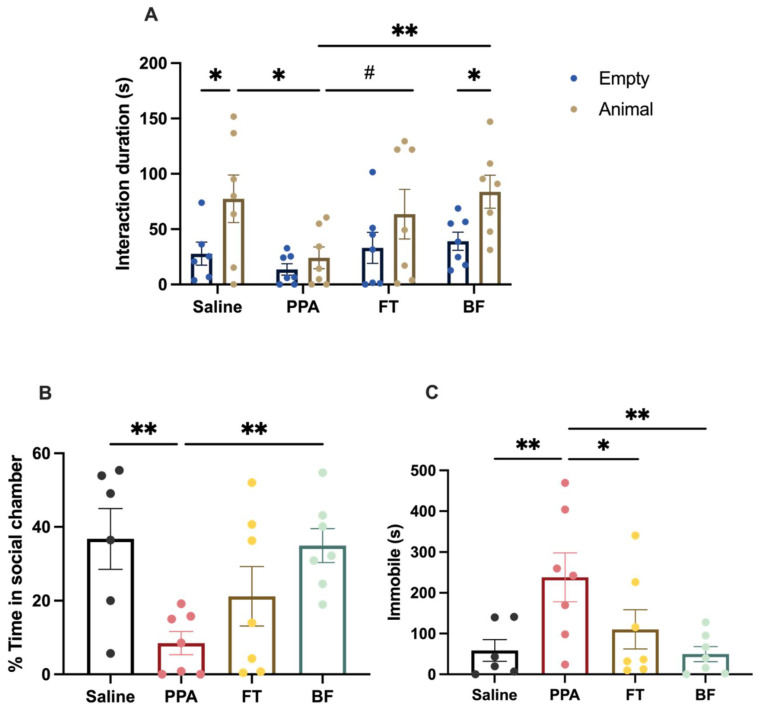
Social interaction in the three-chamber social test following saline (*n* = 6) and propionic acid (PPA) (*n* = 7) treatments, and fecal transplant (FT) (*n* = 7) and Bifidobacterium (BF) interventions (*n* = 7): (**A**) time spent by focal animals interacting with the conspecific animal and the empty holding box during the three-chamber social test; (**B**) percentage of time spent in the social chamber relative to other chambers; (**C**) time spent immobile during the test. Data presented are means ± standard error. Significant two-way and one-way ANOVAs were followed by multiple comparisons by Fisher’s least significance difference ** *p* ≤ 0.01, * *p* ≤ 0.05, # *p* = 0.06.

**Figure 3 brainsci-11-01038-f003:**
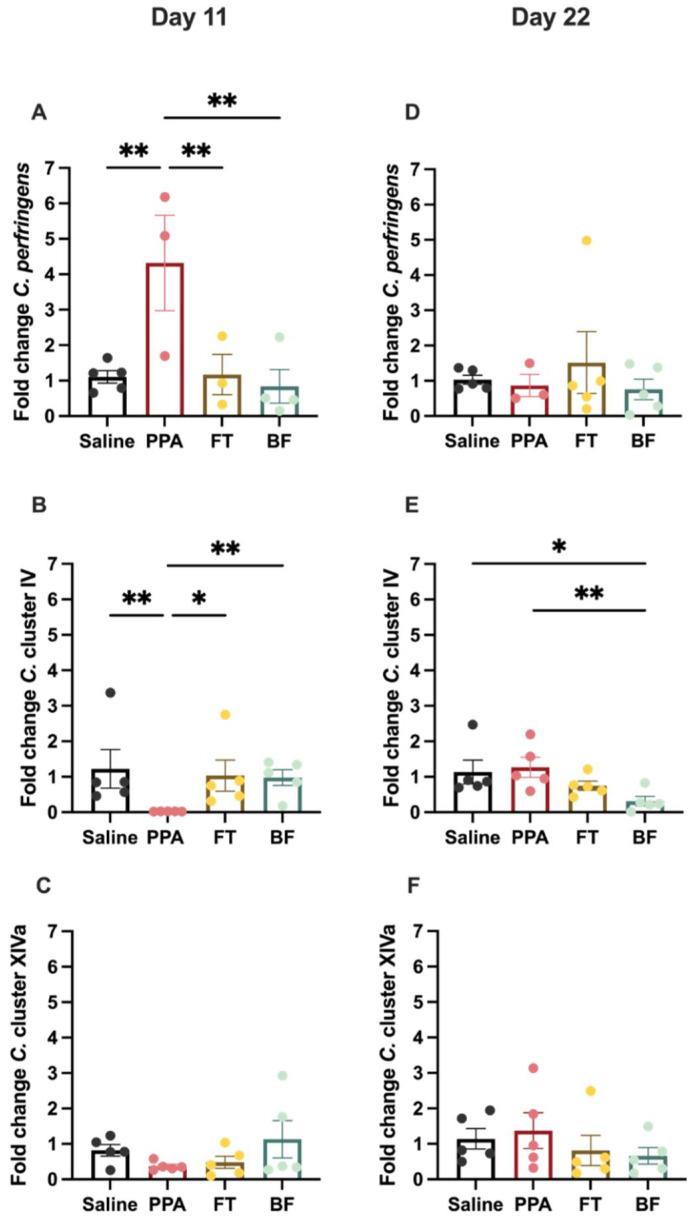
Levels of fecal Clostridium bacteria following saline (*n* = 5) and propionic acid (PPA) (*n* = 3–5) treatments, and fecal transplant (FT) (*n* = 3–5) and Bifidobacterium (BF) (*n* = 4–5) interventions at day 11 and at the last day of the intervention (day 22): (**A**) *Clostridium perfringens*; (**B**) *Clostridium* cluster IV; (**C**) *Clostridium* cluster XIVa at 11 days; (**D**) *Clostridium perfringens*; (**E**) *Clostridium* cluster IV; (**F**) *Clostridium* cluster XIVa at 22 days. Data presented are means ± standard error of 2^−ΔΔCt^ values. Significant one-way ANOVA was followed by multiple comparisons by Fisher’s least significance difference or Kruskal–Wallis followed by uncorrected Dunn’s test. ** *p* ≤ 0.01, * *p* ≤ 0.05.

**Figure 4 brainsci-11-01038-f004:**
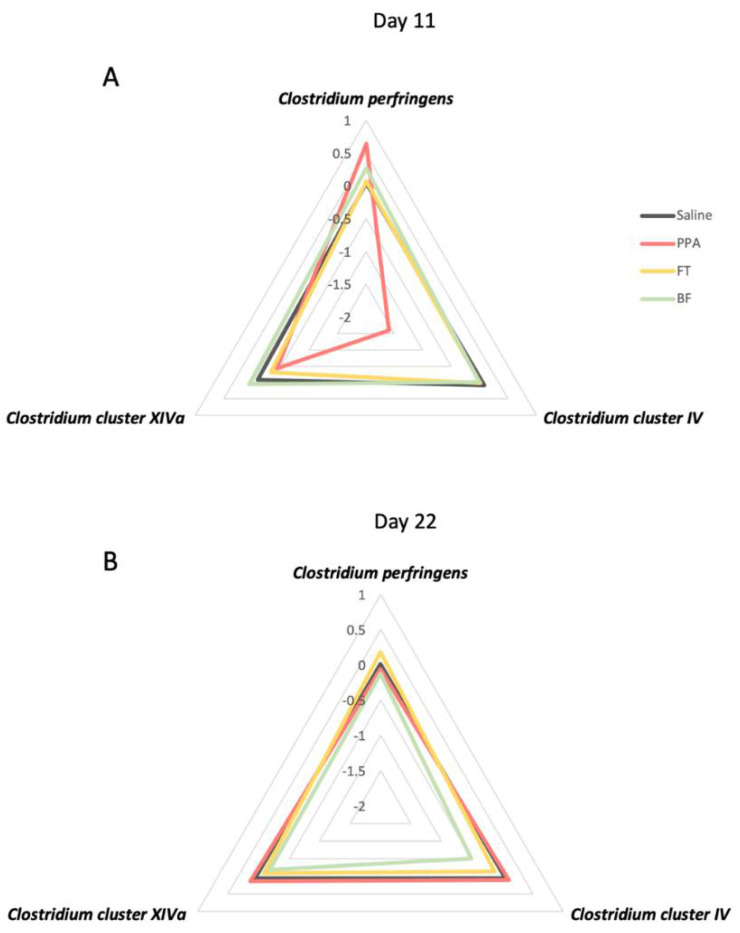
Radar plots of the patterns of fecal *Clostridium* bacteria levels following propionic acid (PPA) treatment and fecal transplant (FT) and *Bifidobacterium* (BF) interventions at day 11 and 22 of the intervention: (**A**) levels of *Clostridium perfringens, Clostridium* cluster IV, and *Clostridium* cluster XIVa at 11 days; (**B**) levels of *Clostridium perfringens, Clostridium* cluster IV, and *Clostridium* cluster XIVa at 22 days. Data presented as log-transformed means of 2^−ΔΔCt^ values.

**Figure 5 brainsci-11-01038-f005:**
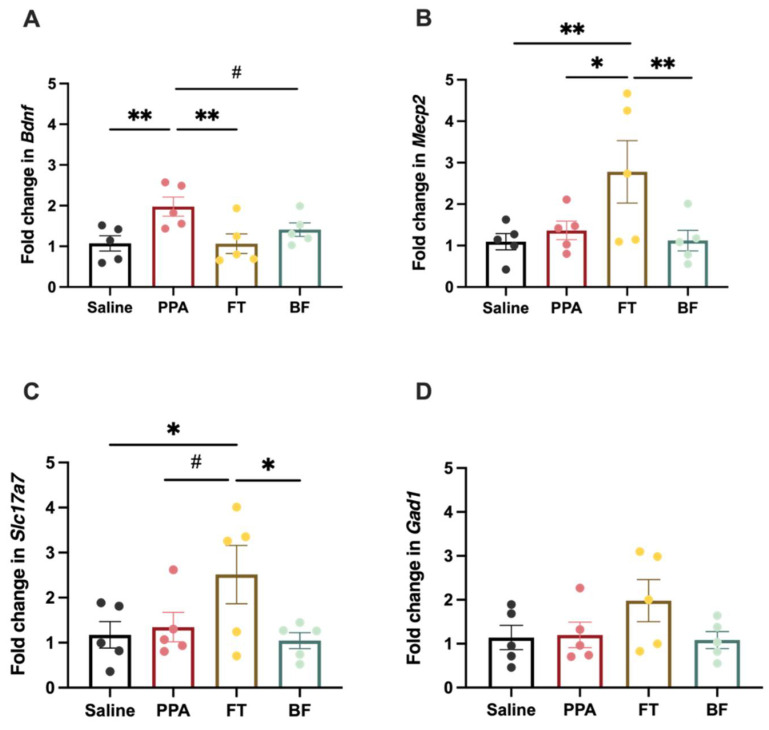
Fold change in hippocampal transcript levels following saline (*n* = 5) and propionic acid (PPA) (*n* = 5) treatments, and fecal transplant (FT) (*n* = 5) and *Bifidobacterium* (BF) (*n* = 5) interventions: (**A**) brain-derived neurotrophic factor (*Bdnf*); (**B**) methyl-CpG binding protein 2 (*Mecp2*); (**C**) solute carrier family 17, member 7 (*Slc17a7*); (**D**) glutamate decarboxylase 1 (*Gad1*). Data presented are means ± standard error of 2^−ΔΔCt^ values. Significant one-way ANOVA was followed by multiple comparisons by Fisher’s least significance difference ** *p* ≤ 0.01, * *p* ≤ 0.05, # *p* = 0.06.

**Table 1 brainsci-11-01038-t001:** List of primer sequences.

Gene	Forward Primer	Reverse Primer
**Fecal Targets**		
Universal Bacterial *16S rRNA* [39]	5′-GTGSTGCAYGGYTGTCGTC-3′	5′-ACGTCRTCCMCACCTTCCTC-3′
*Clostridium perfringens* [40]	5′-GGGGGTTTCAACACCTCC-3′	5′-GCAAGGGATGTCAAGTGT-3′
*Clostridium* cluster IV [41]	5′-GACGCCGCGTGAAGGA-3′	5′-AGCCCCAGCCTTTCACATC-3′
*Clostridium* cluster XIVa [41]	5′-GACGCCGCGTGAAGGA-3′	5′-AGCCCCAGCCTTTCACATC-3′
**Brain Targets**		
*ActinB*	5′-TTTGAGACCTTCAACACCCC-3′	5′-CTGCTGCCTTCCTTGGATG-3′
*18S rRNA*	5’-ATGGTAGTCGCCGTGCCTA-3’	5’-CTGCTGCCTTCCTTGGATG-3′
*Gapdh*	5′-ACATCAAATGGGGTGATGCT-3′	5′-GTGGTTCACACCCATCACAA-3′
*Bdnf*	5′-AAAACCATAAGGACGCGGACTT-3′	5′-AAAGAGCAGAGGAGGCTCCAA-3′
*Mecp2*	5′-CAAACAGCGACGTTCCATCA-3′	5′-TGTTTAAGCTTTCGCGTCCAA-3′
*Slc17a7*	5′-CCTTAGAACGGAGTCGGCTG-3′	5′-AAGATCCCGAAGCTGCCATA-3′
*Gad1*	5’-ACTGGGCCTGAAGATCTGTG-3’	5’-CCGTTCTTAGCTGGAAGCAG-3’

## Data Availability

The data presented in this study are available on request from the corresponding author.

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
