# Peer review of "Fecal Transplant and Bifidobacterium Treatments Modulate Gut Clostridium Bacteria and Rescue Social Impairment and Hippocampal BDNF Expression in a Rodent Model of Autism"

_brainsci, 2021, doi:10.3390/brainsci11081038_

Round 1

Reviewer 1 Report

In the current manuscript, Abuaish et al. explored the implementation of two different interventions that target the microbiota in a rodent model of autism and their effects on social behavior, the levels of different fecal Clostridium bacteria, and hippocampal transcript levels. This study adds onto interesting ongoing work on the role of gut microbiota in autism. Modulation of the gut microbiota is suggested to improve autistic symptoms. The authors induced autism was induced in young male rats using oral gavage of propionic acid (PPA). Although this work is interesting, yet it is not quite polished in order to be published in Brain Science at least in its current state. There are queries that need clarification.

In Fig. 1A, it seems as if PPA treatment reduces interaction time both for empty containers and conspecifics. Why does PPA treatment affect interaction time for empty containers? Did the authors also perform statistics between empty saline vs empty PPA? The FT dataset for conspecific interaction duration seems to have a wide distribution, especially those clustered around 120 seconds are the culprits. The authors should perform outlier analyses to understand if these three datapoints could be included. Alternatively, the authors should increase animal numbers for the behavioral experiments. No. of animals used across each group should be mentioned in the text/figure legends.

Figure 5 poses considerable confusion in interpreting the data presented. Firstly, I am not very clear what BDNF signaling has to do with social behaviors since this is the context in which the BDNF data is presented and interpreted (see discussion). The authors should explicitly state this in their discussion. Secondly, why is Mecp2 selectively increased for FT interventions? Thirdly, what is the authors’ interpretation of the data presented in panels C and D - Do the authors imply that gut microbiota selectively enhances glutamatergic signaling for FT interventions while keeping GABA signaling unchanged, thereby tipping the balance towards more excitation?

Reviewer 2 Report

ASD is a complex neurodevelopmental disorder with each patient exhibiting slightly different manifestations of symptoms across the three major domains: social deficits, language/communication deficits, and repetitive behaviors. There are no treatments for ASD. Some drugs are prescribed off-label to reduce individual symptoms of the disorder. Use of these drugs are controversial and their efficacy questionable. Random use can show significant side effects that outweigh the benefits of the drugs. Though it is difficult, studying ASD is extremely important in order to develop new and better treatments for patients suffering from this often debilitating disorder as well as their families. This study focuses on elucidating the mechanis involved in fecal transplantation and probiotic treatment in microbiota in a rodent model of autism and their effects on social behavior. Since, impaired social interaction is key feature of several major psychiatric disorders including depression, autism and schizophrenia, this study is relevant.

Key Concerns:

  1. For the three chamber social test, was the examiner blinded to the treatment group?
  2. Since the authors have pointed in the discussion part that, fecal transplant has anti-anxiety effect. An open field test is recommended for this research study, to measure anxiety-like and locomotor behaviour.
  3. (BDNF), a key molecule involved in neuronal survival, differentiation, and synaptic plasticity. Signaling by BDNF depends on the proteolytic cleavage of a pro-form of BDNF (proBDNF) to the mature form BDNF, and over the last several years, it has been learned that these two proteins can have very different functions in the brain. For example, proBDNF has been linked to long-term depression and to bind preferentially to the tumor necrosis factor superfamily receptor p75NTR, mediating impairments in neuronal plasticity, apoptosis and neuronal death. Conversely, mature BDNF binds to TrkB and stimulates downstream signaling pathways leading to multiple neurotrophic effects: neuronal differentiation, neurite outgrowth, neuronal survival, strengthening of synapses, neurogenesis, long-term potentiation, and learning and memory. So, in this study what is the role of proBDNF (precursor form of BDNF) in ASD? Did they check the levels of proBDNF. Since. An imbalance in ratio of proBDNF/BDNF will change the outcome of the results.
  4. Display of protein levels would be informative (western blot). Because they checked only the transcript levels and not protein itself. Since, change in expression levels of genes, can change the protein dynamics and subsequently alter the signalling pathways in brain which leads to neuroplasticity. Protein data is considered much stronger than mRNA levels since protein levels show functional changes in the molecules.
  5. In the study, authors looked at BDNF, MeCp2 mRNA levels in hippocampus. Why did they only choose hippocampus? This should be discussed clearly in the paper.
  6. Did they observe any change in prefrontal cortex transcript levels?
  7. Were the brains perfused, since blood will change the outcome?
  8. Can the authors be confident that the changes in transcript levels they identify can be linked to the changes in behaviour?
  9. It would have been useful to explore correlations between the various biochemical measures in order to buttress the argument that they are related. Still, the present results lay a foundation for informative future studies.
  1. I think that the Discussion could be shortened and better focused.
